# Preliminary paleohistological observations of the StW 573 ('Little Foot') skull

Amélie Beaudet[1,2,3,4]*, Robert C Atwood[5], Winfried Kockelmann[6], Vincent Fernandez[7], Thomas Connolley[5], Nghia Trong Vo[5], Ronald Clarke[8], Dominic Stratford[2]

[1]Department of Archaeology, University of Cambridge, Cambridge, United Kingdom; [2]School of Geography, Archaeology and Environmental Studies, University of the Witwatersrand, Johannesburg, South Africa; [3]Institut Català de Paleontologia Miquel Crusafont, Universitat Autònoma de Barcelona, Barcelona, Spain; [4]Department of Anatomy, University of Pretoria, Pretoria, South Africa; [5]Diamond Light Source Ltd, Harwell Science and Innovation Campus, Didcot, United Kingdom; [6]STFC-Rutherford Appleton Laboratory, ISIS Facility, Harwell, United Kingdom; [7]Core Research Laboratories, Natural History Museum, Cromwell Rd, South Kensington, London, United Kingdom; [8]Evolutionary Studies Institute, University of the Witwatersrand, Johannesburg, South Africa

**Abstract** Numerous aspects of early hominin biology remain debated or simply unknown. However, recent developments in high-resolution imaging techniques have opened new avenues in the field of paleoanthropology. More specifically, X-ray synchrotron-based analytical imaging techniques have the potential to provide crucial details on the ontogeny, physiology, biomechanics, and biological identity of fossil specimens. Here we present preliminary results of our X-ray synchrotron-based investigation of the skull of the 3.67-million-year-old *Australopithecus* specimen StW 573 ('Little Foot') at the I12 beamline of the Diamond Light Source (United Kingdom). Besides showing fine details of the enamel (i.e., hypoplasias) and cementum (i.e., incremental lines), as well as of the cranial bone microarchitecture (e.g., diploic channels), our synchrotron-based investigation reveals for the first time the 3D spatial organization of the Haversian systems in the mandibular symphysis of an early hominin.

*For correspondence:
aab88@cam.ac.uk

Competing interests: The authors declare that no competing interests exist.

## Introduction

Applications of X-ray synchrotron-based analytical techniques in evolutionary studies have opened up new avenues in the field of paleoanthropology. In particular, X-ray synchrotron microtomography has been proved to be particularly useful for imaging anatomical structures in extant and fossil hominins that are traditionally observed through destructive histological methods (e.g., *Tafforeau and Smith, 2008*; *Maggiano et al., 2016*; *Andronowski et al., 2017*; *Mani-Caplazi et al., 2017*; *Gunz et al., 2020*). For instance, microscopic analyses of fossil craniodental specimens using synchrotron radiation have revealed previously unknown aspects of the ontogeny of extinct hominin taxa (*Tafforeau and Smith, 2008*; *Gunz et al., 2020*).

Besides its geological age of 3.67 million years, StW 573 ('Little Foot') is remarkable for its outstanding degree of preservation and completeness (*Clarke and Kuman, 2019*). The high-resolution virtual exploration of the 'Little Foot' skull is thus expected to provide new insights into the Pliocene *Australopithecus*' biology. Here, we present preliminary results of our X-ray synchrotron-based investigation of the dentition and cranial bones of 'Little Foot'. The main aim of our study is to identify and assess the degree of preservation of craniodental microstructures that could contribute to the

reconstruction of *Australopithecus*' biology. To the best of our knowledge, this is the first time that histological features of the compact bone of a Pliocene hominin skull have been non-invasively observed.

## Results

### Histology of dental tissues

*Figure 1* shows two dimensional (2D) sections through the roots of the upper right first molar with a resolution of 3.25 µm. These sections reveal the presence of cementum between the dentine and sediments filling the tooth alveolus (*Figure 1A,B*). The dentine–cementum junction and the boundary between the cementum and the sediments are clearly visible. Cementum microstructures, such as incremental lines, are discernible (*Figure 1B*). Additionally, we reconstructed in three dimensions (3D) the enamel cap of the lower left canine using image stacks at 21.23 and 7.91 µm resolution (*Figure 2*). Lines and pits observable on the distal-buccal aspect of the 21.23-µm reconstruction are

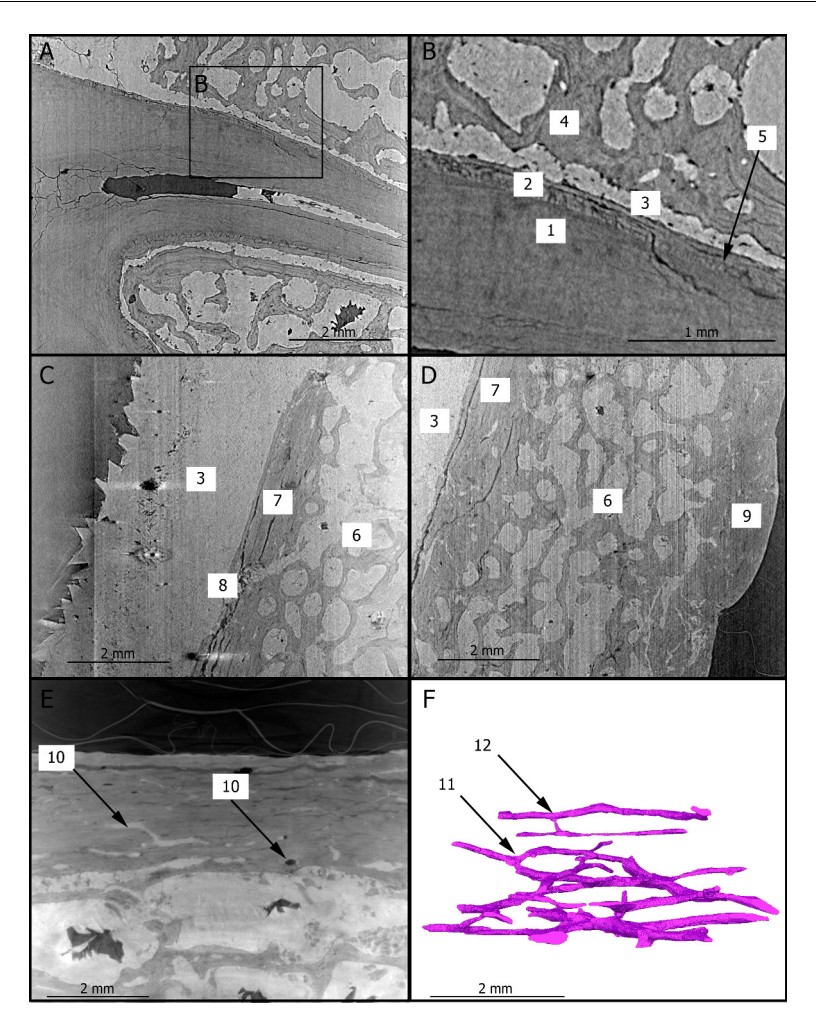

**Figure 1.** 2D sections of the roots of the upper right first molar (A,B), of the cranial vault (C,D), and of the mandibular symphysis (E) of StW 573. The close up (B) of the roots of the upper right third premolar shows the dentine–cementum junction. The Haversian canals of the mandibular symphysis is reconstructed in 3D (F) in the same orientation as in (E). 1: dentine; 2: cementum; 3: calcite; 4: trabecular bone; 5: incremental line in the cementum; 6: diploic bone; 7: inner table; 8: diploic channel; 9: outer table; 10: Haversian canals; 11: dichotomous branching; 12: transverse connection.

**Figure 2.** 3D renderings of the buccal aspect of the enamel surface of the lower left canine of StW 573 using synchrotron-based data sets at 21.23 (A) and 7.91 µm (B). The black arrows (A) point to enamel defects. 3D reconstructions are superimposed (C) and the distances between them are rendered by a pseudo-color scale ranging from dark blue (lowest values) to red (highest values).

located at about 2.4 and 3.6 mm from the cemento-enamel junction, which represents 36% and 54% of the crown height, respectively. While fine details of the pits are clearly visible in the 3D reconstruction using synchrotron-based data sets at 7.91 µm, the color map reveals that they are slightly less clearly rendered in the 21.23-µm 3D reconstruction, especially on the mesial-buccal aspect (*Figure 2C,D*). These reductions in the normal thickness of enamel correspond to disruptions to the normal growth of enamel (i.e., hypoplasias) and indicate two disruptive events in StW 573's life history.

## Histology of bone tissues

*Figure 1C–E* show 2D sections through the cranial vault and the mandibular symphysis with a resolution of 3.25 µm. The voids in the spongious bone have been partially or completely filled by calcite. The opening of the diploic channel through the inner table in the cranial vault could be identified (*Figure 1C*). The Haversian system is discernible in the outer and inner tables of the cranial vault (*Figure 1C,D*) and clearly visible in the compact bone of the mandible (*Figure 1E*). *Figure 1F* shows a 3D reconstruction of the canal network and branching and interconnections between the Haversian systems. Based on the terminology defined by *Maggiano et al., 2016*, we identify a Type 2 (i.e., dichotomous) branching pattern. Furthermore, a linear transverse connection (i.e., Volkmann's canal) can be observed. Vascular canals are proportionally more abundant close to the trabecular bone than in the rest of the compact bone (*Figure 1F*). The Haversian canals globally lie parallel to the external surface.

## Discussion

Collectively, our results show that the virtual histological investigation of both dental and bone tissues in complete fossil hominin skulls by using high-resolution synchrotron radiation may be possible. Contrary to traditional histological analyses based on physical sections of the bone (e.g., *Bartsiokas, 2002*) and imaging-based exploration of volumes of interest physically extracted from bones (e.g., *Maggiano et al., 2016*; *Andronowski et al., 2017*), this method, as previously demonstrated in landmark papers for dental microstructures using dentognathic remains (e.g., *Tafforeau and Smith, 2008*), offers the possibility of non-invasively investigating microscopic structures in complete crania that have deep implications for the ontogeny, physiology, biomechanics, and biological identity of fossil specimens.

More specifically, our observations reveal that 'Little Foot' preserves fine cementum microstructures that can be quantitatively explored. Because cementum is rarely remodeled during life, it may preserve valuable information about hominin paleobiology (rev. in *Tang et al., 2016*). In particular, the incremental lines may be used to determine age-at-death as well as specific stress periods that might be related to life-history events (e.g., pregnancies) or diseases (rev. in *Tang et al., 2016*). Moreover, our high-resolution synchrotron images of the skull reveal the presence of enamel defects in 'Little Foot's' teeth. Enamel hypoplasias are indicators of physiological stress experienced during childhood and may be related to diseases or dietary deficiency/nutritional stress (rev. in *Guatelli-Steinberg and Huffman, 2012*). Interestingly, enamel defects in StW 573's lower left canine are

found at the same distance of the cemento-enamel junction as in other *Australopithecus* lower canines from Sterkfontein (*Guatelli-Steinberg, 2003*). Our study thus further confirms that high-resolution synchrotron radiation may reveal very fine details of enamel defects, with substantial differences in the appreciation of hypoplasia between 21.23 and 7.91 microns of spatial resolution.

Additionally, the opening of the diploic vessels in the cranial vault of 'Little Foot' could be imaged. Since this structure may be involved in brain thermoregulation, future synchrotron-based 3D virtual reconstruction of the diploic network in early hominin skulls could be particularly useful for determining when and how the complex human-like thermal regulation system emerged (rev. in *Bruner, 2017*). In particular, such data would contribute to explore the potential correlations between large proportions of diploic bone in *Australopithecus* and related expansion of the diploic vessels (*Beaudet et al., 2018*).

To the best of our knowledge, this is the first time that histological features of the compact bone of a Pliocene hominin skull have been non-destructively observed. Since the microscopic organization of the compact bone may have age-related biological significance, the presence, identification, and characterization of such histological structures are of particular interest for determining the age-at-death of fossil specimens (e.g., *Ericksen, 1991*), particularly in combination with the dental markers discussed above. For example, the branching patterns identified in this study are also found in extant humans (*Maggiano et al., 2016*). In the sample analyzed here, we could only identify one transverse connection, which confirms the advanced age of StW 573 since this type of branching is more common in young individuals (*Maggiano et al., 2016*; *Clarke and Kuman, 2019*). Similarly, the spatial organization of the vascular canal network has been suggested to reflect functional adaptation throughout the individual's life and/or growth rate in fossil long bones (*Ricqlès et al., 2000*). Consequently, this histomorphological parameter represents a relevant proxy for evaluating the properties of the loading environment or developmental pattern in fossil specimens (*Ricqlès et al., 2000*). In our preliminary results, the vascular canals are more abundant close to the trabecular bone in the mandible of StW 573, which may indicate an area of intense bone remodeling, potentially in response of biomechanical loading. The fact that the vascular canals could be successfully reconstructed in 3D in the mandible of a 3.67-million-year-old fossil specimen such as 'Little Foot' reveals the invaluable contribution of synchrotron radiation in refining our knowledge of fossil hominin paleobiology at a histological level. For instance, a more comprehensive analysis of the compact bone microstructures should provide new insights into the evolution of the bone modeling/remodeling process, which is a fundamental aspect of bone functional adaptations in the human lineage. We might expect that, because of changes in the biomechanical environments (in the masticatory system but also in relation with locomotory adaptations), the organization of the Haversian system in the skeleton may have varied throughout the hominin lineage.

## Materials and methods

### X-ray synchrotron microtomography

We performed propagation phase-contrast synchrotron X-ray micro-computed tomography (PPC SXCT) at the I12 beamline of the Diamond Light Source, United Kingdom (*Drakopoulos et al., 2015*). We used two setups to (1) image the full skull with an isotropic voxel size of 21.23 µm and (2) image regions of interest with an isotropic voxel size of 7.91 and 3.25 µm. PPC SXCT of the full skull was performed using the External Hutch Two of I12, providing the largest beam allowing for imaging the whole skull at once. The X-ray beam was set to a monochromatic energy of 140 keV (double bent Laue Si 111 monochromator). At the level of the detector, the beam size was 75.84 mm horizontally and 19.28 mm vertically. Projections were recorded using the large field of view indirect detector from I12 comprising a cadmium tungstate scintillator, 0.3× magnification optical lenses and two PCO.edge 5.5 sCMOS cameras (PCO AG, Kelheim, Germany). Complete imaging of the specimen consisted of 21 individual acquisitions, moving the specimen vertically each time by 10 mm, to use the brightest part of the beam providing an overlap of slightly over 50%, which was used later to increase the signal to noise ratio. The centre of rotation was shifted near the edge of the projection, effectively doubling the reconstructed field of view, by reflecting and assembling the complementary image sections as described. Each acquisition consisted of 9000 projections of 0.25 s each over a 360° rotation of the sample. Additionally, 50 flatfield images (sample out of the beam) were

recorded before and after the series of acquisition as well as 10 dark images (X-ray beam off to record the noise of the camera). The regions of interest were scanned in I12 Experimental Hutch One using the beamline's modular imaging system. This indirect detector consists of a PCO. edge 5.5 sCMOS camera and four user-selectable optical modules, each comprising a scintillator, 90-degree turning mirrors, and a visible light lens. We used modules 2 and 3 with a magnification of 0.820× and 2×, respectively, corresponding to a recorded pixel size of 7.91 µm and 3.25 µm, respectively.

## Reconstruction

The full-skull images taken with the large field of view camera were reconstructed using prototype processing scripts implemented in Python and using the ASTRA library for the tomographic reconstruction step. The radiographs were pre-processed using X-ray noise removal, reference image ratio (flatfield correction), ring artifact suppression (*Vo et al., 2018*), and a low-pass phase-propagation-based filter (after *Paganin et al., 2002*). Tomographic reconstruction used the Filtered Back-Projection algorithm on a graphical processing unit (GPU) via the ASTRA library (*Ramachandran and Lakshminarayanan, 1971*; *van Aarle et al., 2016*). The image data for the whole skull consisted of 21 individual, overlapping, vertical segments. Each segment was pre-processed and reconstructed with fine adjustment of the overlap determined by slice-by-slice comparison in the reconstructed images. Due to synchrotron beam loss during the pre-programmed scan, two segments had to be re-scanned subsequently resulting in small additional variations in the contrast for that segment.

The region-of-interest tomographs were reconstructed using the SAVU tomographic processing software (*Atwood et al., 2015*; *Wadeson and Basham, 2016*) developed at Diamond Light Source. In the reconstruction process, ringartifact removal (*Vo et al., 2018*) and autocentering (*Vo et al., 2014*) were applied, as well as distortion correction as necessary (*Vo et al., 2015*). The low-pass filter approach (*Paganin et al., 2002*) was applied, and both filtered and unfiltered reconstructions were evaluated; the filtered output exhibits increased visible contrast between materials and decreased noise, but suffers some blurring of boundaries. Filtered back-projection reconstructions were performed using the Astra library (*Ramachandran and Lakshminarayanan, 1971*; *van Aarle et al., 2016*), and the whole process was applied on an HPC cluster system using the SAVU tomography pipeline.

## 2D and 3D observations

We focused our preliminary qualitative observations on the microstructural organization of the teeth and of the compact bone of the cranial vault and the mandible. We thus selected 2D sections from the high-resolution (i.e., 3.25 µm) X-ray volumes documenting structures of interest in (1) the roots of the upper right first molar, (2) the parietal eminence, and (3) the mandibular symphysis (*Figure 1*). Moreover, we reconstructed in 3D (1) the enamel surface of the lower left canine (at 21.23 and 7.91 µm; *Figure 2*) and (2) the vascular canals preserved in the mandibular symphysis (at 3 µm, *Figure 1F*) using the segmentation tools available in Avizo v9.0 (Visualization Sciences Group Inc). 3D reconstructions of the enamel cap of the lower left canine using the 21.23 and 7.91 µm scans were automatically aligned, and the distances between the two were mapped onto the enamel surface using the Avizo modules 'Align surfaces' and 'Surface distance' (*Beaudet et al., 2018*).

## Acknowledgements

This work was carried out with the support of the Diamond Light Source, instrument I12 (experiment MG21334-1). We are indebted to B Zipfel (Johannesburg) for the loan of the StW 573 skull. For scientific discussion/collaboration we are grateful to: F de Beer (Pelindaba), L Bruxelles (Nîmes), K Carlson (Los Angeles), J Heaton (Birmingham), K Jakata (Johannesburg), T Jashashvili (Los Angeles), K Kuman (Johannesburg), A Le Cabec (Bordeaux), T Pickering (Madison).We are grateful to the University of the Witwatersrand for loaning StW 573. Permission for fossil access was granted by B. Zipfel. For their financial support, we would like to thank the Department of Science and Innovation (DST), the DST-NRF Center of Excellence in Palaeosciences (CoE-Pal), the National Research Foundation (NRF), the National Research Foundation African Origins Platform, the Palaeontological Scientific Trust (PAST), the University of the Witwatersrand and the Diamond Light Source and the ISIS facility of the Science and Technology Facilities Council (STFC). We thank the editors and the two

reviewers for their comments, which contributed to improving the original version of this manuscript. Opinions expressed and conclusions arrived at are those of the authors and are not necessarily to be attributed to the Center of Excellence in Palaeosciences.

## Additional information

### Funding

| Funder | Grant reference number | Author |
|---|---|---|
| DST-NRF Centre of Excellence in Palaeosciences | | Amélie Beaudet |
| National Research Foundation | African Origins Platform AOP180616347591 | Dominic Stratford |
| National Research Foundation | Equipment-related Travel and Training Grant ART180724350809 | Dominic Stratford |
| University of the Witwatersrand | University Research Committee | Dominic Stratford |
| Palaeontological Scientific Trust | | Dominic Stratford |

The funders had no role in study design, data collection and interpretation, or the decision to submit the work for publication.

### Author contributions

Amélie Beaudet, Conceptualization, Data curation, Formal analysis, Funding acquisition, Validation, Investigation, Visualization, Methodology, Writing - original draft, Project administration, Writing - review and editing; Robert C Atwood, Conceptualization, Resources, Data curation, Software, Visualization, Methodology, Writing - original draft, Writing - review and editing; Winfried Kockelmann, Conceptualization, Methodology, Writing - review and editing; Vincent Fernandez, Conceptualization, Investigation, Methodology, Writing - original draft, Writing - review and editing; Thomas Connolley, Conceptualization, Data curation, Methodology, Writing - original draft, Project administration, Writing - review and editing; Nghia Trong Vo, Software, Investigation, Methodology, Writing - original draft, Writing - review and editing; Ronald Clarke, Data curation, Writing - original draft, Writing - review and editing; Dominic Stratford, Conceptualization, Funding acquisition, Writing - original draft, Project administration, Writing - review and editing

### Author ORCIDs

Amélie Beaudet https://orcid.org/0000-0002-9363-5966
Robert C Atwood https://orcid.org/0000-0001-6708-1085
Winfried Kockelmann https://orcid.org/0000-0003-2325-5076
Vincent Fernandez https://orcid.org/0000-0002-8315-1458
Thomas Connolley https://orcid.org/0000-0002-1851-3467
Nghia Trong Vo https://orcid.org/0000-0002-3683-7377
Ronald Clarke https://orcid.org/0000-0003-1759-8937
Dominic Stratford https://orcid.org/0000-0001-9790-8848

### Decision letter and Author response

Decision letter https://doi.org/10.7554/eLife.64804.sa1
Author response https://doi.org/10.7554/eLife.64804.sa2

## Additional files

### Data availability

All data generated or analysed during this study are included in the manuscript and supporting files.

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
