## [Decision Letter]

**Acceptance summary:**

This works provides interesting new insights into the biology of human ancestors. Further the imaging techniques used permit non-destructive studies on fossil specimens and permits comparison to present day humans. In fact there is significant preservation of structure and function in this fossil specimen. The data further provides some insights into the stresses and hardships in the environment in which these early humans lived.

**Decision letter after peer review:**

Thank you for submitting your article "Preliminary palaeohistological observations of the StW 573 ('Little Foot') skull" for consideration by *eLife*. Your article has been reviewed by two peer reviewers, including Carlos Isales as the Reviewing Editor and Reviewer #1, and the evaluation has been overseen by Mone Zaidi as the Senior Editor. The following individual involved in review of your submission has agreed to reveal their identity: Jack Yu (Reviewer #3).

Summary:

This is a brief report of paleohistology, using micro CT, to examine the cranium and maxillary dento-alveolar complex (enamel, dentine, cementum and alveolar bone). The resolution was 3.25 micrometer, the specimen was fossilized Pliocene hominin, *Australopihecus* StW 573, "Little Foot", from 3.67 million years ago. The images, mostly 2D with some 3D renditions, were of high quality , reconstructed using the Filtered Back-Projection Algorithm. This technical paper has considerable strengths. First, it confirms the ability of microCT to visualize histological features of fossilized specimens, such as the trabecular architecture, Haversian system, and dento-alveolar complex. Second, it demonstrates the detection of incremental lines in the cementum which correlate well with age of the individual. Finally, the article documents empanel hypoplasia, at the same distance from the cemento-enamel junction from another *Australopithecus* mandibular canines.

Essential Revisions:

1) The method, though fascinating, is not really new. The first Guatelli-Steinberg reference, for example, is from 2003. Because of that, simply introducing the methodology is not sufficient and the manuscript should contain other novel findings and their potential implications. Please discuss this aspect.

2) The author(s) pointed out some intriguing features but did not give any biological significance.

3) This paper is not the first time that histological features of compact bones were examined using microCT on Pliocene hominin skulls. 3.67 million years separate Homo sapient from *Australopithecus*. There are, no doubt, differences at the genetic and gross anatomical levels. What are the expected changes in histology, in any?

4) It would be helpful for the authors to provide more information on the physiological relevance of their findings and also how it compares to present day bone and tooth structure.

---

## [Author Response]

Essential Revisions:1) The method, though fascinating, is not really new. The first Guatelli-Steinberg reference, for example, is from 2003.

We agree that this analysis of hypoplasia in fossil hominins in general, and in *Australopithecus* in particular, is not new. However, the idea of this paper is to insist on the fact that scanning a fossil hominin skull using the synchrotron technology not only provides information about the gross anatomy, but also about very fine structures, from increments in dental tissues (as illustrated in previous papers) to the cranial vascular system (as demonstrated here for the first time). The comparison with Guatelli-Steinberg, 2003, aims to reinforce the idea that this kind of information is crucial for the continuing development of the field of palaeobiology. Moreover, we did emphasis in the Introduction (and mentioned this again in the beginning of the Discussion in the revised version) that this technology has been previously used in palaeontology, but that we go further by imaging fine details of the compact bone from an entire cranium.

Because of that, simply introducing the methodology is not sufficient and the manuscript should contain other novel findings and their potential implications. Please discuss this aspect.

In the Discussion, we synthetized the main findings (increments of the cementum, hypoplasia, diploic vessels and Haversian system) and, for each of them, detailed biological implications. Moreover, we would like to emphasize that this is the first time that such tiny structural details of the compact bone could be reconstructed in 3D in a Pliocene hominin skull. This represents incredible potential for improved palaeohistological perspectives. We understand that this aspect should be further elaborated and have thus added further relevant information in the last paragraph of the Discussion. We also added a sentence in the introduction clarifying the aim of the paper.

2) The author(s) pointed out some intriguing features but did not give any biological significance.

In the Discussion, we did present the biological significance of each feature explored in the study. In the second paragraph of the Discussion, we explained that increments in the cementum could be used to determine the age at death and stress periods that could be related to life-history traits. We also discussed the presence of hypoplasias, that could indicate important physiological stress such as disease or nutritional stress. In the third paragraph, we explained that the diploic system likely contributes to brain thermoregulation and, as such, could help us understand hominin brain evolution. The last paragraph is dedicated to the Haversian system. We added further information in the revised version so that the biological implications of this finding (i.e., bone modeling/remodeling) could be better emphasized.

3) This paper is not the first time that histological features of compact bones were examined using microCT on Pliocene hominin skulls.

There were indeed previous studies applying synchrotron radiation to fossil hominin craniodental specimens (e.g., Carlson et al., 2011 Science; Le Cabec et al., 2015 PLoS One; Smith et al., 2015 PLoS One; Gunz et al., 2020), focusing on dental histological details and brain imprints. However, to the best of our knowledge, this is the first time the compact bone is investigated using synchrotron radiation in a Pliocene hominin skull. If not, we would be extremely grateful if the reviewer could provide relevant references.

3.67 million years separate Homo sapient from *Australopithecus*. There are, no doubt, differences at the genetic and gross anatomical levels. What are the expected changes in histology, in any?

We absolutely agree that changes happened in the hominin lineage. In that respect, our experiment demonstrates that even microscopic changes in the biology of fossil hominins could be followed. Concerning the dental structures explored here, hypoplasias will indeed vary depending on the physical and social environments. The organization of the diploic vessels surely evolved with the brain. Changes in the Haversian system are more difficult to anticipate, as there is little information available at the moment about the organization of the vascular system at this scale in fossil hominins. We added a sentence regarding this aspect at the end of the Discussion.